# Athlete’s Passport: Prevention of Infections, Inflammations, Injuries and Cardiovascular Diseases

**DOI:** 10.3390/jcm9082540

**Published:** 2020-08-06

**Authors:** Cristina Mennitti, Mariarita Brancaccio, Luca Gentile, Annaluisa Ranieri, Daniela Terracciano, Michele Cennamo, Evelina La Civita, Antonietta Liotti, Giovanni D’Alicandro, Cristina Mazzaccara, Giulia Frisso, Raffaela Pero, Barbara Lombardo, Olga Scudiero

**Affiliations:** 1Department of Molecular Medicine and Medical Biotechnology, University of Naples Federico II, Via S. Pansini 5, 80131 Naples, Italy; cristinamennitti@libero.it (C.M.); cristina.mazzaccara@unina.it (C.M.); giulia.frisso@unina.it (G.F.); 2Department of Biology and Evolution of Marine Organisms, Stazione Zoologica Anton Dohrn, Villa Comunale, 80121 Naples, Italy; mariarita.brancaccio@szn.it; 3Ceinge Biotecnologie Avanzate S. C. a R. L., 80131 Naples, Italy; gentilelu@ceinge.unina.it (L.G.); ranieria@ceinge.unina.it (A.R.); 4Department of Translational Medical Sciences, University of Naples “Federico II”, 80131 Naples, Italy; daniela.terracciano@unina.it (D.T.); michelecennamo@unina.it (M.C.); evelina.lacivita@unina.it (E.L.C.); tonialiotti@virgilio.it (A.L.); 5Department of Neuroscience and Rehabilitation, Center of Sports Medicine and Disability, AORN, Santobono-Pausillipon, 80122 Naples, Italy; ninodalicandro@libero.it; 6Task Force on Microbiome Studies, University of Naples Federico II, 80100 Naples, Italy

**Keywords:** sports medicine, elite athletes, serum biomarkers, prevention, infection, inflammation, muscle injuries, cardiovascular diseases

## Abstract

Laboratory medicine in sports medicine is taking on an ever-greater role in the assessment and monitoring of an athlete’s health condition. The acute or intense exercise practiced by elite athletes can lead to the appearance of infections, inflammations, muscle injuries or cardiovascular disorders, whose diagnosis is not always rapid and efficient, as there is no continuous monitoring of the athlete. The absence of such monitoring can have serious consequences in terms of recovery of the professional athlete. These imbalances can induce metabolic adaptations which translate into alterations of specific parameters in terms of concentration and activity. The aim of this study was to follow the variation of specific biochemical biomarkers in a basketball team participating to the maximum championship during different phases of the agonistic season. The evaluation of serum biomarkers can help doctors to safeguard the athlete’s health and sports trainers to adapt workouts, thus avoiding the appearance of diseases and injuries that in some cases can be underestimated by becoming irreversible ailments that do not allow the athlete to return to a healthy state. This information can be useful to create athlete biologic passports.

## 1. Introduction

Laboratory medicine in sport has the task of monitoring the athlete’s health through specific laboratory tests to improve athletic performance, reduce the risks associated with the intense physical effort to prevent the appearance of infections, inflammations, muscle injuries or cardiovascular disorders. Moreover, personalized follow-up allows the athlete to return to competition [1,2]. In particular, the use of biochemical and hematological screening tests and a correct interpretation of the parameters obtained is essential for laboratory biologists, sports doctors and coaches to identify risk factors in athletes early [1,3].

Exercise, intense training and sports competitions can induce changes in the serum concentrations of numerous laboratory parameters [4]. Enzymes such as creatine kinase (CK) and lactate dehydrogenase (LDH), which regulate muscle metabolism, are generally increased after exercise [5,6,7,8,9]. A CK increase represents a marker of cell necrosis and acute or chronic muscle damage [5], in which the serum concentration of CK increases proportionally with exercise and tissue damage [3]. In particular, the increased CK is closely related to the exercise’s duration rather than its intensity [6]. In athletes undergoing intense training, CK usually increases after 24–36 h, making it an essential biomarker for monitoring recovery efficiency [7].

In the same way as CK, an increase in LDH levels has been observed after intense exercise and muscle diseases. For this reason, it can be used as a marker of muscle damage [7]. LDH is an enzyme present in the form of five different isoenzymes (LDH1-5) expressed in various tissues, with a prevalence of isoenzyme-5 (LDH5) at the muscle level [3]. Unlike CK, these isoenzymes have shown different trends in power and endurance athletes, allowing them to evaluate metabolic adaptation to exercise.

Moreover, exercise is also a potent endocrine stimulator that causes changes in hormonal concentrations that depend on various factors, including intensity, duration, training mode and training status of the subject [10,11]. The hormone most involved in an athlete’s clinical evaluation is cortisol, a catabolic hormone produced by the adrenal gland. In skeletal muscle, cortisol regulates homeostasis and energy metabolism [12,13]. Several authors have shown that exercise causes an increase in cortisol levels caused by the intensity of the training and the pre and post-competition emotional stress [14,15,16,17].

Another noteworthy factor for the clinical evaluation of an athlete is vitamin D. The latter, as well as being essential for homeostasis and bone metabolism, has direct effects on the muscle [18]. Vitamin D is mainly produced on the skin through a chemical reaction that depends on sunlight exposure and partly taken with the diet. In athletes, the blood levels of vitamin D vary and depend on the geographical position, ethnicity, local climatic conditions and sports disciplines (indoor vs. outdoor) [19,20,21]. Defects in vitamin D production lead to bone mineralization and muscle function defects [21]. For this reason, sports doctors and nutritionists must regularly monitor the concentration of vitamin D [22,23,24].

Athletic performance can also be affected by changes in thyroid hormone levels [25,26]. Hypothyroidism can cause a variety of symptoms in athletes, including fatigue, reduced energy and endurance, weight gain and reduced athletic performance, while hyperthyroidism is responsible for insomnia, fatigue, rapid weight loss, muscle weakness and loss of muscle mass. Thyroid hormone levels can increase, decrease or remain unchanged depending on the type of exercise, intensity and duration [3]. These ambiguous results could be due to various elements such as the athlete’s nutritional status, environmental factors and the blood sampling procedure.

Another essential aspect of the athlete’s clinical evaluation is the analysis of the red and white components of the blood through a complete blood count (CBC). During intense exercise, the oxygen demand in the skeletal muscle increases. To compensate for this increase in demand, through increased stimulation of erythropoiesis, red blood cells (RBC) increase in number, but their average corpuscular volume (MCV) decreases [27]. This compensation does not cause alterations in the hematocrit value (Hct). In some athletes, it is common to find reduced hemoglobin (Hb) and hematocrit, in the absence of pathologic situations. This common condition during intense physical activities has been called “sports anemia.” The various forms of sports anemia include: (i) a condition known as pseudoanemia: endurance athletes tend to have slightly low Hb levels according to general population standards because aerobic exercise expands the basic plasma volume and dilutes the level of Hb, despite no change in the red blood cell mass [28]. This diluting pseudoanemia cannot be considered as damage to the athlete [29,30]; (ii) an anemic form characterized by a reduction in the concentration of Hb and Hct, due to a drastic accentuation of intravascular hemolysis. This phenomenon causes an increase in free Hb, which can cause Hb loss in the urine (hemoglobinuria); (iii) an iron deficiency anemia (the most common) [31]: since iron is necessary for oxygen transport and energy metabolism, in endurance athletes with iron deficiency, there can be a reduction in exercise capacity and VO2 max, the maximum amount of oxygen that the body can use [32].

Indeed, high-intensity exercise increases iron losses by up to 70% compared to sedentary populations.

Exercise is sometimes responsible for the onset of cardiovascular disease (CVD) including coronary artery disease and cerebrovascular disease [33,34].

Several studies have shown that acute exercise causes an increase in platelet count (PLT) due to hemoconcentration and platelet release from the liver, lungs and spleen. Lippi et al. showed that PLT and MPV (mean platelet volume) increased significantly after the run and returned to baseline after three hours [35]. Another study evaluated the same parameters and came to the same conclusion: an increase follows resistance exercise in the aggregation of PLT, PCT (plateletcrit), an MPV and platelets [36].

Finally, the C-reactive protein (CRP) represents another interesting biochemical parameter for athletes’ physical evaluation. CRP is a detectable protein in the blood produced by the liver and part of the so-called acute-phase proteins, a group of proteins synthesized during an inflammatory state [37]. Its measurement, together with erythrocyte sedimentation rate (ESR), can be instrumental in case of suspicion of inflammatory states of infectious origin and some inflammatory diseases such as rheumatoid arthritis. Moreover, CRP is used to evaluate the presence and extent of the phlogistic response. Monitoring its levels helps determine disease progress and therapeutic efficacy. Furthermore, damage to the walls of the vessels can be caused by infiltration into the cell wall and inflammatory elements, for this reason, CRP levels can be used to establish the level of risk of cardiovascular disease [38,39] and its increase could also highlight the presence of genetic pathologies [39,40,41,42,43].

Our study aimed to dose and evaluate specific parameters such as enzymes of muscle metabolism, hormonal markers and hematological parameters to analyze elite athletes’ health. Monitoring these parameters will help customize their training, competition and recovery regimes, optimizing performance and also could help in the realization of an athlete’s passport.

## 2. Experimental Section

### 2.1. Ethical Approval

The study was conducted in accordance with the ethical guidelines of Helsinki Declaration of the World Medical Association and was approved by ethics committee (protocol 200/17) of the School of Medicine, University of Naples Federico II.

### 2.2. Participants

For this study, we recruited twelve professional basketball players. All subjects were informed about the purpose and procedures of the study and written informed consent was obtained from each participant. The physical characteristics of the players as mean (±SD) were age years 27 ± 7 years, weight 88 kg ± 10 kg, height 193 cm ± 6 cm. None of the subjects smoked, drank alcohol or consumed drugs known to alter chemical parameters of the blood exam completed or hormonal formula. All subjects followed a similar diet throughout the season and, above all, the same diet during the study, which was monitored continuously by the team doctors. Every day the calorie intake is around 3000 kcal, in specific: carbohydrates about 55–60% of the daily calorie intake, protein about 12–15% of the daily calorie intake, against the 10–12% recommended for those who do not play sports, total lipids 25–30% of the daily calorie intake, minerals according to the recommendations valid for the population general (LARN, recommended levels of energy and nutrient intake for the Italian population—Italian Society of Human Nutrition ‘96), vitamins according to the recommendations valid for the general population (LARN), water at least 1–1.5 L per day and in any case as much as is lost through sweat, urine, etc. The players followed the same training program, i.e., they trained every day for two sessions, a morning session consisting of a gym workout for 2 h and an afternoon session consisting of basketball practice for three hours. This training program was followed daily, except on the days of official games played during the season (two games per week).

Each game had a duration of 40 min, the 40 min are divided into 4 periods of 10 min each. The athletes included in the study are all men, with a competitive activity of 7 years. The athletes included in the study had no previous muscle injuries, cardiovascular disorders or recurring infections.

### 2.3. Experimental Approach

The collection of blood samples from athletes has been performed in three different phases of the agonistic season. The first blood samples were collected in September (0 months) in the preseason phase. Then, samples were taken on November (1 month after the start of the championship) and in February (3 months after the start of the championship). A panel of biochemical and hematological markers was applied to the blood samples of the athletes. For this study, those involved in muscle metabolism, endocrine system and hematological parameters capable of providing a general picture of the athlete’s health will be taken into consideration. Thus, blood samples were handled with appropriate laboratory techniques to evaluate the following parameters: creatine kinase (CK), lactate dehydrogenase (LDH), thyroid hormone (fT3 (triiodothyronine), fT4 (thyroxine), TSH (thyroid stimulating hormone)), CRP, cortisol, vitamin D, red blood cells and platelets indices. Blood samples were taken in the morning (8:00 a.m.) before training for all athletes, after 72 h of rest. Each athlete was subjected to a blood sample, the waiting time was about 10 min. The measurement was performed in duplicate.

### 2.4. Procedures

Red blood cells and platelet counts were performed through the Siemens Advia 2120i hematology analyzer. Serum cortisol concentration was determined by immunoassay procedures through the Immulite 2000 analyzer (Cortisol Immunoassay kit; Siemens Healthiness, Erlangen, Germany). The dosage of CK, LDH and CRP was evaluated on Architect c16000 (creatine kinase, lactate dehydrogenase and C-reactive protein assay; Abbott Diagnostics, Chicago, IL, USA). The dosage of vitamin D and thyroid hormones (fT3, fT4 and TSH) was performed using automated enzyme immunoassays in chemiluminescence using Liaison XL (Diasorin) and (ADvia Centaur, Siemens Heltineers, Erlangen, Germany), respectively. All the procedures took place according to the manufacturer’s recommendation.

### 2.5. Statistical Analysis

All statistical analyses were performed using the GraphPad Prism 8.4.0 software (GraphPad Software, Inc., La Jolla, CA, USA). Data were expressed as the mean  ±  standard deviation. Student’s *t*-test made comparisons among groups. Values of *p* < 0.05 were considered significant. To evaluate the relationships between cortisol/FT3, cortisol/FT4 and cortisol/TSH the Pearson linear correlation coefficient was used, which has a value between −1 and +1, where +1 corresponds to the perfect positive linear correlation, 0 corresponds to an absence of linear correlation and −1 corresponds to the perfect negative linear correlation.

## 3. Results

### 3.1. Effect of Exercise on Red Blood Cells

To highlight how prolonged exercise can affect the health of professional athletes, we evaluated red blood cells (Figure 1A–D). It is possible to note that erythrocytes, hematocrit and hemoglobin have no significant variations (Figure 1A–C). Moreover, we verified an increase in the average corpuscular volume (MCV) (Figure 1D). In particular, the MCV increases between 1 and 3 months in comparison to 0 months (Figure 1D).

### 3.2. Exercise and Platelet Indices

To determine how intense exercise can influence the trend of platelet indices we analyzed first the platelets (Figure 2A)—and in this case, no significant variation was observed (Figure 2A); we can note the same trend for plateletcrit (Figure 2B). Finally, the trend of the average platelet volume (MPV) is different; in fact, there is a moderate increase at 1 and 3 months compared to 0 months (Figure 2C).

### 3.3. Dosage of CK, LDH, Vitamin D and Cortisol

To reveal how physical activity can influence the trend of some biochemical parameters essential for the correct muscle metabolism, we have dosed CK, LDH, vitamin D and cortisol (Figure 3A–D). CK levels have no significant variation between 0 and 3 months (Figure 3A). LHD levels undergo a slight decrease if we compare months 0 with months 3 (Figure 3B). Moreover, also the levels of vitamin D decrease at 1 month and 3 months if compared with 0 months (Figure 3C). In addition, the decrease of its levels is significant also compared 1 month with 3 months (Figure 3C). Finally, cortisol level levels have no significant variation between 0 and 3 months (Figure 3D).

### 3.4. The Impact of Exercise on the Thyroid Gland

To understand how physical activity can influence the trend of thyroid hormones, we carried out a dosage of triiodothyronine (fT3), thyroxine (fT4) and thyroid-stimulating hormone TSH. From this analysis, it is possible to see that there is no significant variation of the three hormones (Figure 4A–C). However, there is a moderate increase in the FT4 hormone at 1 month when compared to month 0 (Figure 4B).

### 3.5. Comparison between Cortisol and Thyroid Hormones

To shed light on how professional physical activity can influence the metabolism of the hormones, we evaluated three parameters. In this case, we focused our attention on the cortisol/FT3 ratio (Table 1); on the cortisol/fT4 ratio (Table 1) and the cortisol/TSH ratio (Table 1). To emphasize the trend over time of these ratios, we used the *Pearson* linear correlation coefficient. From the analysis conducted, it emerges that the cortisol/fT3 ratio has a negative linear correlation in months 0 and 3 (Table 1), at 1 month we can see the absence of correlation (Table 1). Other fascinating data can be seen from the cortisol/fT4 ratio; in fact, the two hormones have a negative linear trend at 0 months (Table 1) at 1 months we can observe a moderate positive correlation (Table 1), instead of at 3 months, we cannot note a correlation (Table 1). For cortisol/TSH ratio, we note a positive correlation in months 0 and 3 (Table 1), at 1 month we can see an absence of correlation (Table 1).

### 3.6. C-Reactive Protein and Inflammation in Athletes

To clarify how physical exercise can influence the athlete’s inflammatory status, we evaluated serum CRP levels. From the data obtained, we find a significant increase in three months from the start of the competitive season (Figure 5).

## 4. Discussion

In recent years, laboratory medicine has taken on a central role in sports and in monitoring the athlete’s health [3]. Numerous scientific studies have shown that intense and prolonged exercise can cause metabolic adaptations, particularly in the serum concentrations of numerous biochemical parameters. The variations of these parameters represent the changes that occur in the body in response to the intensity and duration of training exercises, as well as the stress to which the different muscle units involved in athletic performance are subjected. These adaptations translate into alterations of specific parameters, in terms of concentration and activity, and their identification could represent a new monitoring method for the health of the athlete. We have focused our study on changes in biochemical, serologic and hematological parameters in professional athletes. Therefore, we evaluated several parameters since the response to exercise-induced stress implies a complex involvement of organs and tissues. For the purposes of our study, we subjected a group of professional basketball athletes to a peripheral blood sample.

Following the observation of the results obtained, we observed that the erythrocytes, hematocrit and hemoglobin did not show significant variations [28]; however, we have noticed a significant decrease in the average corpuscular volume (MCV). These data allow us to sustain that in our case an intense physical exercise can cause the appearance of anemia. In particular, the decrease in MCV could lead to the appearance of iron deficiency anemia. The symptoms of iron deficiency, caused by an albeit slight decrease in MCV are weakness, fatigue, decreased exercise performance, increased heart rate, respiratory fatigue, headache and dizziness [31]. For this reason, monitoring these parameters during the competitive season can be of major relevance.

Simultaneously, with the evaluation of the erythrocyte indices, we evaluated the platelet component. In this case, the athletes showed significant variations neither in the number of platelets nor in the platelet count;, but a variation in the mean platelet volume is noted. Recent studies have shown that after physical exertion, MPV can increase [33,34,35,36]. The monitoring of this parameter together with the evaluation of the number of platelets and the platelet count can be relevant to establish cardiovascular or thrombotic risks in the athlete [33,34,35,36].

In our study, the athletes examined had not had previous manifestations of cardiac or vascular disorders; consequently, the increase in the MPV could be caused by fatigue due to both training and competitions.

Then, we decided to evaluate two important markers for muscle metabolism: CK and LDH. Our study has shown that both parameters slightly decrease. Several studies have pointed out that patients with persistently high CK activity show a simultaneous alteration of LDH profiles. Therefore, monitoring of CK and LDH concentrations is necessary to evaluate the athlete’s muscle function and response, therapies and recovery times in case of muscle damage [1,2,3,4,5,6,7]. Our athletes show reduced levels between zero and three months and the absence of persistent muscle damage.

Together to these parameters, we also evaluated the CRP level. The dosage of serum CRP levels is of great importance for the evaluation of infectious and inflammatory diseases. It is known that competitive athletes are often affected by respiratory diseases or skin infections [43,44]. In addition, cardiovascular diseases can also be monitored through CRP [37,38,39,42,43,44,45,46]. Our data show that athletes have an increase in CRP values with an increase in physical activity. This increase is probably due, on one hand, to possible past infections that have not been efficiently eradicated; on the other hand, to the rupture of the vessels with consequent appearance of phlogistic events. Therefore, the dosage of this marker associated with the evaluation of the platelet component can help the doctor to better evaluate the risk of thrombotic events, while the single evaluation can give may represent an alarm bell for the presence of bacterial infections that often affect the athlete [37,38,39,42,43,44,45,46,47,48,49,50,51,52,53,54,55,56,57,58,59].

Finally, we evaluated the hormonal component, by measuring the levels of vitamin D, cortisol and thyroid hormones (fT3, fT4 and TSH). Athletes have shown a decrease in vitamin D levels. Low serum vitamin D levels are associated with high rates of muscle and bone injuries; therefore it is important that the team of doctors, including nutritionists, plan a diet that allows a sufficient intake of vitamin D so that this deficiency does not affect the athlete’s performance and health [18,19,20,21,22,23,60]. As for the dosage of Cortisol and that of thyroid hormones, we have pointed out how the trend of these hormones is closely related. An increase in cortisol is known to block the synthesis of thyroid hormones.

In our case, cortisol levels do not show a significant variation, but if compared with thyroid hormones (see Table 1) in particular with fT3, they show an inversely related trend at months zero and three. With fT4 only at months zero is observed a slight inversely related trend, while if compared with the TSH they have a parallel almost trend. Monitoring cortisol and thyroid hormones is crucial for two reasons; one because cortisol is involved in muscle metabolism, so its levels indicate muscle health [12,13,14]; second, thyroid hormones can influence the athlete’s health [61,62], an incorrect synthesis can cause hypo or hyperthyroidism; in both cases, these pathologies can have serious causes if not properly treated. This is why doctors must pay more attention to the hormonal component of each athlete.

Basketball is a team sport that stimulates all parts of the body, promoting harmonious muscle development. It is an alternate aerobic–anaerobic physical activity, in which moments of walking/running, short shots and jumps follow one another. After football and volleyball, basketball occupies third place in the ranking among the most practiced sports in Italy: eight hundred thousand people who practice it according to Istat; half of which are members of FIP (Italian Basketball Federation). While an excellent activity for children and teenagers, when one has exceeded the threshold of 40 one must be more careful, because the risk of injury increases, especially if one dos not have a good athletic preparation. Running, passing and shooting for goals tone different body muscles, both in the upper and lower parts. Maintaining a good balance—the “core” (abdominal, buttocks and lumbar)—is also stressed. The calories that can be burned during an hour of play (depending on the intensity and one’s weight) are 600–900—a result that contributes to maintaining a correct weight. However, this effort can lead to situations of poor hydration, fatigue and weakness. Basketball is a very dynamic sport, with rapid and unpredictable game phases and for this reason it also helps to develop reflexes and coordination. The results obtained therefore reflect the advantages and disadvantages of this sport. In this scenario, our study indicates that during intense and prolonged physical activity, the human body puts in place a series of defense and adaptation mechanisms to contrast a prolonged inflammatory state that could be harmful to the athlete. Therefore, an extension of laboratory investigations must be carried out to assess the physical state of athletes by ensuring constant monitoring to achieve a maximum level of prevention in sport.

## 5. Conclusions

In our study, we highlighted that some biochemical parameters vary between the preseason period and the third months from the beginning of the championship in a group of professional basketball athletes. In particular, we have observed a decrease in LDH and vitamin D and at the same time, an increase in MPV and PCR. A decrease in MCV was also found in the athletes. These findings have shown that the continuous monitoring of an athlete allows to intervene promptly and develop adequate training and nutritional programs in order to prevent conditions such as anemia. Furthermore, the cortisol/fT3 ratio shows an inverse proportionality at zero and three months. The biochemical parameters analyzed in this work could be used as indices to evaluate and monitor the physical condition of an athlete at different stages of the sports session. This approach can guarantee the protection of the athlete’s health and also, prevent any loss of shape or the occurrence of any cardiovascular diseases, muscle injuries and metabolic diseases. These data suggest that athletes need constant monitoring to avoid the emergence of severe pathologies.

In conclusion, this study can lay the foundation for design a panel of specific biomarkers, which can be applied to the individual athlete. The results obtained could be reported in a personalized biologic passport for each athlete that would allow sports doctors and athletic trainers, respectively to program personalized workouts, nutrition and therapies.

## Figures and Tables

**Figure 1 jcm-09-02540-f001:**
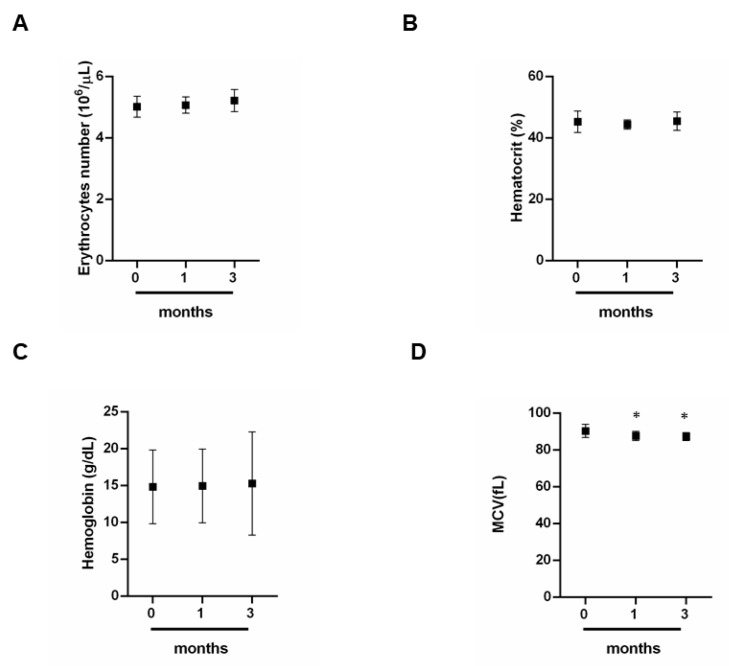
Red blood cells. (**A**–**D**) Assessment of percentages of red blood cells at different times 0, 1 and 3 months by a group of 12 professional basketball athletes. Data expressed as the mean ± SD. Significance determined by the *t-*test student; (**D**) 0 months versus 1 months * (*p* = 0.0400) and 0 months versus 3 months * (*p* = 0.0152). MCV, average corpuscular volume.

**Figure 2 jcm-09-02540-f002:**
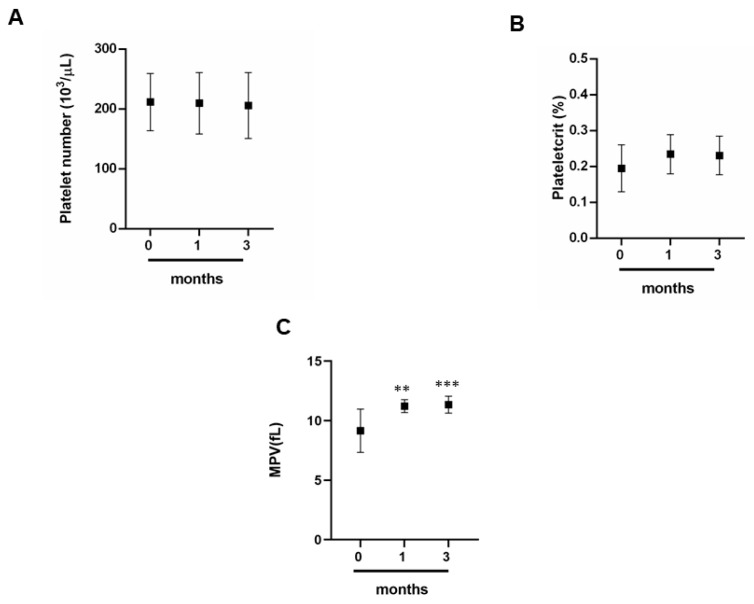
Platelet indices. (**A**–**C**) Assessment of percentages of red blood cells at different times 0, 1 and 3 months by a group of 12 professional basketball athletes. Data expressed as the mean ± SD. Significance determined by the *t-*test student; (**C**) 0 months versus 1 months ** (*p* = 0.0010) and 0 months versus 3 months *** (*p* = 0.008). MPV, mean platelet volume.

**Figure 3 jcm-09-02540-f003:**
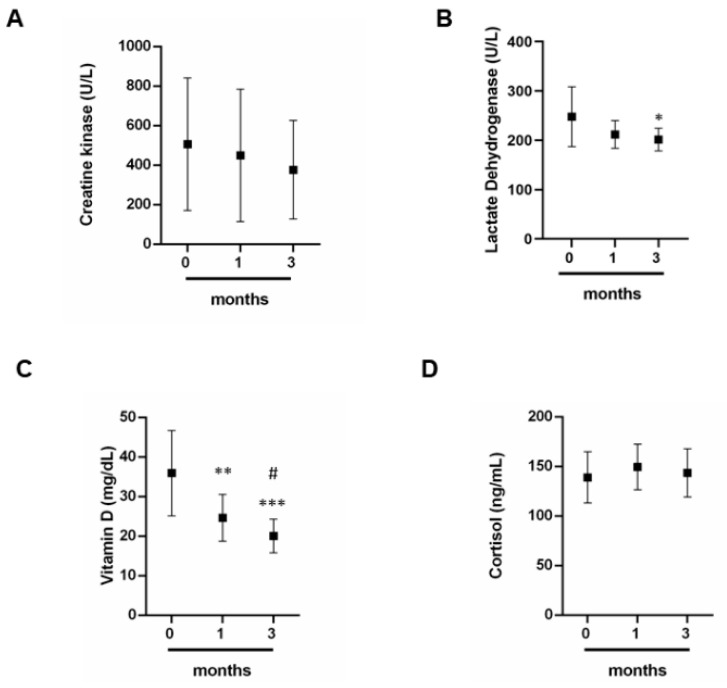
Biochemical parameters. (**A**) Creatine kinase, (**B**) LDH, (**C**) vitamin D and (**D**) cortisol dosage in professional athletes at different times 0, 1 and 3 months since the start of the championship by a group of 12 professional basketball athletes. Data expressed as the mean ± SD. Significance determined by the *t-*test student; (**B**) 0 months versus 3 months * (*p* = 0.0219); (**C**) 0 months versus 1 months ** (*p* =0.0043); **0** months versus 3 months *** (*p* = 0.001) and 1 months versus 3 months # (*p* = 0.0398). LDH, lactate dehydrogenase.

**Figure 4 jcm-09-02540-f004:**
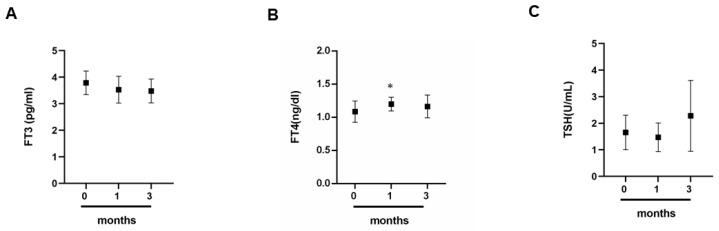
Hormonal dosage. (**A**) fT3 (**B**) fT4 (**C**) TSH evaluation of thyroid hormones in professional athletes at different times 0, 1 and 3 months by a group of 12 professional basketball athletes. Data expressed as the mean ±SD. Significance determined by the *t-*test student; (**B**) 0 months versus 1 months * (*p* = 0.0481). fT3, triiodothyronine; fT4, thyroxine; TSH, thyroid stimulating hormone.

**Figure 5 jcm-09-02540-f005:**
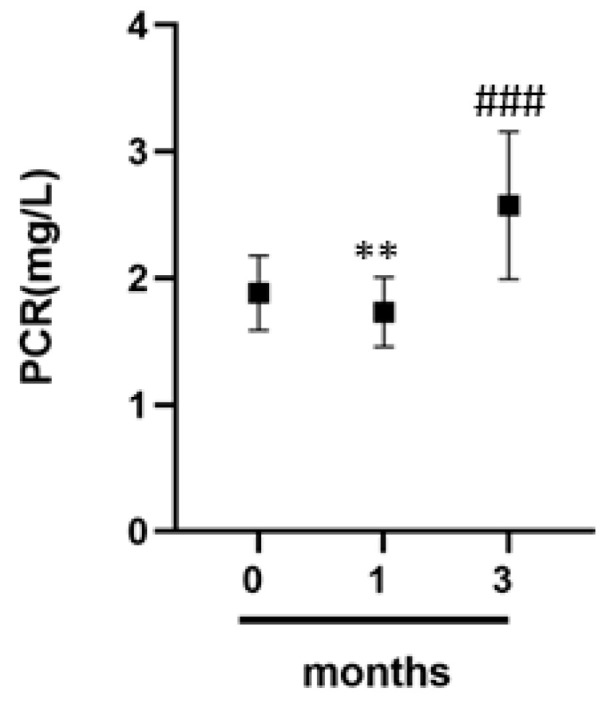
C-reactive protein level. C-reactive protein (CRP) dosage in professional athletes at different times 0, 1 and 3 months by a group of professional basketball athletes. Data expressed as the mean ± SD. Significance was determined by the *t*-test student: 0 months versus 3 months ** (*p* = 0.0014) and 1 months versus 3 months ### (*p* = 0.0002).

**Table 1 jcm-09-02540-t001:** Pearson’s correlation (***ρ***) between cortisol/fT3 ratio (B) cortisol/fT4 ratio (C) cortisol/TSH ratio. Pearson index was considered significant for values between +1 and −1.

Variables	*ρ*		
	0 Month	1 Month	3 Months
Cortisol/fT3	−0.46	0.08	−0.13
Cortisol/fT4	−019	0.20	0.04
Cortisol/TSH	0.11	0.05	0.21

fT3, triiodothyronine; fT4, thyroxine; TSH, thyroid stimulating hormone.

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
