# Peer review of "Athlete’s Passport: Prevention of Infections, Inflammations, Injuries and Cardiovascular Diseases"

_jcm, 2020, doi:10.3390/jcm9082540_

Round 1

Reviewer 1 Report

This study does not present any novelty. Several published studies have already investigated in professional athletes the biomarkers monitored in this work.

The number (n) of athletes tested is low, also, it would be recommended to monitoring athletes practicing different disciplines/sport, therefore subjected to different metabolic activity (e.g. aerobic and anaerobic training).

The authors indicate their approach useful to guarantee the protection of the athlete’s health, for instance, the occurrence of any cardiovascular diseases. However, to better satisfy this aim, the serum PCR levels should be investigated in association with other circulating markers of cardiovascular health, such as ANP, NT-proBNP, or BNP.

The sole serum PCR dosage, to evaluate infection and inflammatory conditions in athletes, does not appear sufficient and selective to guarantee the protection of the athlete’s health. The authors could show other reliable inflammation assays, such as WBC count, neutrophil count, lymphocyte count, neutrophil-lymphocyte ratio, Cytokine Panel. The same considerations can be extended to PCR dosage as a unique indication for the risk of thrombotic events.

Author Response

Point-by-point response.

Reviewer 1

  1. This study does not present any novelty. Several published studies have already investigated in professional athletes the biomarkers monitored in this work.

Response:

Thanks to the Reviewer 1 for the helpful comments.

The novelty of our study consists of monitoring a team of athletes every two months for two consecutive sports sessions. This continuous monitoring has enabled us to identify the most significant variations of specific parameters and, above all, to promptly intervene in the case of abnormalities to prevent accidents and/or drop of shape.

  1. The number (n) of athletes tested is low, also, it would be recommended to monitoring athletes practicing different disciplines/sport, therefore subjected to different metabolic activity (e.g. aerobic and anaerobic training).

Thanks to the Reviewer 1 for the helpful comments.

Response:

The basic idea of our study is to identify the changes of specific biomarkers. For this reason, we have chosen to monitor an unique team of elite athletes (consisting of 12 athletes) and, on their blood samples, we have applied a large panel of biochemical and haematological parameters. By monitoring these athletes and always respecting the same methodologies and withdrawal times, any changes in the parameters were easily identified.

We appreciate the advice of the Reviewer 1 and, for our future studies, we take into consideration the idea to monitor athletes practicing different sports.

  1. The authors indicate their approach useful to guarantee the protection of athlete’s health, for instance, the occurrence of any cardiovascular disease. However, to better satisfy this aim, the serum PCR levels should be investigated in association with other circulating markers of cardiovascular health, such as ANP, NT-proBNP, or BNP.

Thanks to the Reviewer 1 for the helpful comments.

Response:

On athlete’s blood sample we have performed the NT-proBNP dosage. The results have shown that this parameter remains within normal range, and for this reason we have decided to not include in this article. However,  we add a table in which the results are reported such as “value ± standard deviation”.

NT-BNP (normal range: <125 pg/mL)

Athlete

0 month

1 month

3 month

Athlete 1

35 ± 33,40

50 ± 21,56

24 ± 32,10

Athlete 2

118 ± 33,40

70 ± 21,56

44 ± 32,10

Athlete 3

31 ± 33,40

60 ± 21,56

45 ± 32,10

Athlete 4

32 ± 33,40

62 ± 21,56

97 ± 32,10

Athlete 5

45 ± 33,40

92 ± 21,56

35 ± 32,10

Athlete 6

38 ± 33,40

37 ± 21,56

20 ± 32,10

Athlete 7

98 ± 33,40

96 ± 21,56

62 ± 32,10

Athlete 8

45 ± 33,40

91 ± 21,56

106 ± 32,10

Athlete 9

50 ± 20,53

44 ± 32,40

96 ± 31,10

Athlete 10

61 ± 31,10

37 ± 31,40

35 ± 33,10

Athlete 11

43 ± 33,11

96 ± 32,40

21 ± 33,10

Athlete 12

32 ± 33,42

43 ± 32,40

62 ± 31,10

Moreover, it is important underlaying that the athletes undergo a cardiological examination, and in the case of abnormalities, appropriate specialist instrumental test are performed in order to protect the athletes from the risk of cardiovascular disease.

  1. The sole PCR dosage, to evaluate infection and inflammatory conditions in athletes, does not appear sufficient and selective to guarantee the protection of athlete’s health. The authors could show other reliable inflammation assays, such as WBC count, neutrophil count, lymphocyte count, neutrophil-lymphocyte ratio, Cytokine Panel. The same consideration can be extended to PCR dosage as unique indication for the risk of thrombotic events.

Thanks to the Reviewer 1 for the helpful comments.

Response:

On the athletes’ blood samples, we applied a general panel of analytes also including the blood count test. We chose to not include the data in the results since the changes were not statistically significant.

Analyte

0 months

1 months

3 months

leukocytes

5.47±1.01

5.99±1.25

5.80±1.24

lymphocytes

37.44±6.94

35.57±8.71

37.34±7.76

neutrophils

48.86±7.11

53.04±9.25

51.09±8.01

monocytes

6.91±1.76

7.26±1.18

7.22±1.26

eosinophils

3.61±2.07

2.65±1.31

3.65±2.27

basophils

0.76±0.42

0.80±0.32

0.69±0.33

Reviewer 2 Report

  • This study has measured various parameters, which have been discussed previously, as potential effectors of physical performance and injury/disease resilience in athletes. The authors observe the course of these parameters during 3 months of twelve basketball player while their season is underway. While the study design was kept simple, but straight forward, the results are not discussed critically. The authors conclusions are not justified by their results, lack a sophisticated in depth-discussion and are very general without adding valuable novel relevant informations. This study does not seem suitable for publication.

Major issues:

  • The second part in the discussion is highly speculative. The authors found no changes in hematological parameters despite a variation of MCV within the normal range. Their conclusion, that intense physical activity can cause iron deficiency anemia, is not supported by the results. Furthermore, there are numerous studies, that have confirmed such an observation. 
  • The sixth paragraph in the discussion is about the change in CRP, which was significantly higher after 3 months. The authors correctly list several potential causes for this observation. However, simply interviewing the participants at the various measurement points about current physical complaints, like infection or chronic overuse injuries would have provided some answers to their speculation.
  • In paragraph 7 of the discussion, the authors declare the observation of decreasing Vitamin D levels throughout the 3 month study period. However, a critical discussion of this observation needs to involve the season of the year, when the blood samples were obtained. Most likely it was winter, when Basketball seasons are typically under way. At this time of the year, a decrease of vitamin D levels can be observed in many individuals, independent from their level of exercise. Furthermore, a critical discussion of vitamin d levels in athletes demands the consideration of the cut-off issue. The cut-off values in vitamin D levels vary between nutritional societies and have changed considerably over the last decades. This is, because, the correlation of a value with the clinical significance is still unknown. Thus, nobody knows, where the cut-off for a true vitamin d deficiency lies demanding supplementation.

Minor issues:

  • Setting the level of significance at p<0.05, it is unneccessary to differentially label p-values <0.01 and <0.001 in the figures. One asterisk for each test with p<0.05 is sufficient and accurate.
  • Presentation of the mean values measured for the outcome parameters at every time point in a table would improve readability
  • Please change PCR to the common label CRP

Author Response

Reviewer 2

This study has measured various parameters, which have been discuss previously, as potential effectors of physical performance and injury/disease resilience in athletes. The authors observe the course of these parameters during 3 months of twelve basketball players while their season is underway. While the study design was kept simple, but straight forward, the results are not discussed critically. The authors conclusions are not justified by their results, lack a sophisticated in depth-discussion and are very general without adding valuable novel relevant informations. This study does not seem suitable for publication.  

Major issues:

  • The second part in the discussion is highly speculative. The authors found no changes in hematological parameters despite a variation of MCV within the normal range. Their conclusion, that intense physical activity can cause iron deficiency anemia, is not supported by the results. Furthermore, there are numerous studies, that have confirmed such an observation.

Thanks to the Reviewer 2 for the helpful comments.

MCV values ​​are decreased within normal values. This result was made possible thanks to the continuous monitoring of the athletes, since, in case of alterations, personalized training programs are developed in order to protect the health of the athlete.

However, in order to satisfy your request, we have modified the section 5 lines 379-381.

  • The six paragraph in the discussion is about the change in CRP, which was significantly higher after 3 months. The authors correctly list several potential causes for this observation. However, simply interviewing the participants at the various measurement points about current physical complaints, like infection or chronic overuse injuries would have provided some answers to their speculation.

Thanks to the Reviewer 2 for the helpful comments.

  • In the paragraph 7 of the discussion, the authors declare the observation of decreasing Vitamin D levels throughout the 3 month study period. However, a critical discussion of this observation needs to involve the season of the year, when the blood sample were obtained. Most likely it was winter, when basketball seasons are typically under way. At this time of the year, a decrease of vitamin D levels can be observed in many individuals, independent from their level of exercise. Furthermore, a critical discussion of vitamin D levels vary between nutritional societies and have changed considerably over the last decades. This is, because, the correlation of value with the clinical significance is still unknown. Thus, nobody knows, where the cut-off for a true vitamin D deficiency lies demanding supplementation.

Thanks to the Reviewer 2 for the helpful comments.

Response:

The blood levels of vitamin D are variable and depend on several factors such as geographical position, season, ethnicity, local climatic conditions, and sports disciplines (indoor vs. outdoor). An adequate intake of vitamin D will have beneficial effects on athletes by reducing the risk of stress fractures, infectious diseases, and inflammations, allowing to optimize muscle function, remodeling, bone health and athletic performance. In the athletes, the dosage of Vitamin D is indicative to give intake recommendations in order to maintain adequate Vitamin D levels.

The collection of blood samples from athletes took place in three different sessions: on September  (0 months) in the preseason phase; on November (1 month after the start of the championship); on February (3 months after the start of the championship).

 Therefore, we have modified the section 2.3 lines 197-199.

Minor Issue:

  • Setting the levels of significance at p<0.05, it is unnecessary to differentially label p-values <0.01 and 0.001 in the figures. One asterisk for each test with p< is sufficient and accurate.

Respone:

Thanks to the Reviewer 2 for the helpful comments.

Thanks for the advice, but we have labeled used a *(p <0.05) ** (p <0.01) and *** (p <0.001) represent significance compared to 0 months; # (p <0.05), ## (p <0.01) and ### (p <0.001)  represent significance compared to 1 months. In this way, we understand the values that present a greater or lesser significance.

  • Presentation of the mean values measured for the outcome parameters at every time point in a table would improve readability.

 Thanks to the Reviewer 2 for the helpful comments.

        We enclose the raw data below, but we prefer to keep the graphics and text unchanged.

  • Please change PCR to the common label CRP.

Thanks to the Reviewer 2 for the helpful comments.

Response:

In order to satisfy your request, we have now modified section 1 lines 151, 153, 158,160, 161, 165; section 2.3 line 204; section 2.4 line 211; section 3.6 lines 298, 301; section 4 lines 342,345.

Reviewer 3 Report

The aim of the present study was to monitor a number of blood biomarkers during a 3 months training period (at 0, 1- and 3-months’ time points during the season) in basketball athletes. Although the aim of the study is clear and simple, I only understand it when reading the experimental approach in the Method section.

General comments

- The title and the abstract seem related to a literature review paper and not an original paper. Both title and abstract as well as the introduction are too vague.

- Too long introduction (vague, reviewing the effect of acute exercise on different biomarkers) and the method section was small with no enough information. The results section lack accuracy in the text (exact p values, effect size etc.)

- In the abstract the aim was not clear, and there are no methods or results sections.

- Previous study showed an association between weather seasonality and blood parameters. As this study last for 3 months (possible weather seasonality change) the use of a control group is necessary to control this confounding variable.

Introduction

- The rational of this study need to be more clarified. Indeed, this study focused on the chronic effect of training program on a number of biomarkers. However, the majority of cited studies in the introduction were dealing with the acute effect of physical exercises. It would be more appropriate to include previous studies dealing with physiological adaptation to different form of chronic exercises or training program or adaptation in different athletic populations (soccer, individual sports etc.) The rational of using basketball players should also be clear in the introduction.

- The authors have identified a number of biomarkers that can help monitoring health status in athletes as it was previously been showed to be altered during exercises. However, it should be noted that in the majority of these study the post exercise levels did not exceed the normal range in healthy individual and that levels alteration was explained by water loss especially during prolonged exercise.

- The authors indicate in the title and through the manuscript “Athletes’ passport” and they selected some biomarkers to create this passport. However, such biological passports cannot be limited to these biomarkers. A simple example is related to the inclusion of CRP (used by previous study as inflammation biomarker). Many studies showed that CRP level did not significantly change during exercise. It is also suggested to control Hs-CRP with at least one or two myokines (e.g., IL-6) to assess inflammatory responses. Additionally, it is well known that prolonged or high-intensity exercise results in oxidative damage to macromolecules in both blood and skeletal muscle; with overtraining induced high production of ROS may negatively affect skeletal muscle contractile function and muscle fiber adaptation. So, when speaking about an athlete’s biological passports it seems also important to include biomarkers of redox status. Therefore, I suggest avoiding using athletic passport and to make the used vocabulary as simple as possible, especially that the used design and tested parameters did not reflect such title and purpose (Athlete's passport: prevention of infections, 3 inflammations, injuries and cardiovascular diseases)

- Please provide the expansion at the first occurrence of all abbreviation (e.g., MPV, PCT)

Methods

- L197 “blood samples from athletes took place in three different sessions: at 0 months in the preseason phase; 1 month after the start of the championship; 3 months after the start of the championship” what you mean by “0 months in the preseason” is this baseline values just after the long between seasons break?  

Please give more information about the basketball season (length, how many matches, duration between matches etc.) Please argue the choose of these 3 time points. In which micro-cycle did you collect blood (e.g., opening, developing, stabilization, checking, tapering, racing, recovery) and what was the load (volume, intensity) during 2-3days just before collecting blood?

Indeed, collecting blood just the day after a match and collecting blood at the end of a recovery micro-cycle will blind the effect of training period (e.g., 3 months of training). Did the authors ensured similar condition (e.g., amount of physical activities, diet behaviors sleep quantity) during at least 72h preceding the blood sample in each blood sampling time-points?

- Blood parameters showed also to follow a circadian variation, please specify the time of day for all blood samples.

- L193: “The athletes included in the study had no previous muscle injuries, cardiovascular disorders, or 194 recurring infections” what about during the experimental period?

- About participants; please specify the gender, years of experiences, and the sample size calculation.

- Is there a standardized meal the day of each blood sample?

- Blood sample (quantity of blood, position of athlete during the blood sample (e.g., sitting how many minutes he was sitting before blood sample) duplicate analysis or not?). Methods for each biomarker should be provided.

- Statistical analysis, how normality of the data has been tested?

- Blood was taken in 3 time points, why the data was not analyzed using the one way Anova for repeated measures and post-hoc test (of course if it is normally distributed) ?

- Effect size (e.g., d cohen) should be provide with p values.

Results

- When stating “To determine how intense exercise” the reader will think that you are evaluation the acute response following an intense exercise. Therefore, it should be clear through the manuscript that this study evaluates the effect of chronic (or 3 months) basketball training program on some blood parameters.

- Table 1. It is not clear if this table is dedicated to present the correlation between cortisol and fT3, Cortisol and fT4 and Cortisol and TSH at the different time point or this table showed the ration values (e.g., Cortisol/fT3 etc.)

- Exact p values with effect size (d cohen) should be presented in text when analyzing the differences in mean between time points.

Discussion

- L320 “however, we have noticed a significant decrease in the average corpuscular volume (MCV). This data allows us to sustain that in our case an intense physical exercise can cause the appearance of anemia” before taking such conclusion, it should be clarified for all parameters values in all time point if these values are still in the normal (healthy) range or not.

Please avoid speaking about the effect of an intense physical exercise. As I mentioned above, we speak about the effect of intense exercise when comparing pre-exercise with post-exercise values (acute effect). However, in the present study you are collecting blood just before exercises in different time point of the seasons. So, you are looking for adaptation in different season period or effect of chronic or regular basketball training program (3months).

- Please go deeper and try to explain to underlying mechanisms behind the present results and also try to link the results with the specificity of the training program in Basketball. Comparing the observed adaptation following basketball training program with other previously studied discipline will strengthen the discussion.

- What about the results of the correlation, they are absent in the discussion.

Author Response

Reviewer 3

The aim of the present study was to monitor a number of blood biomarkers during a 3 months training period (at 0, 1- and 3-months’ time points during the season) in basketball athletes. Although the aim of the study is clear and simple, I only understand it when reading the experimental approach in the Method section.

General comments

- The title and the abstract seem related to a literature review paper and not an original paper. Both title and abstract as well as the introduction are too vague.

- Too long introduction (vague, reviewing the effect of acute exercise on different biomarkers) and the method section was small with no enough information. The results section lack accuracy in the text (exact p values, effect size etc.)

- In the abstract the aim was not clear, and there are no methods or results sections.

Thanks to the Reviewer 3 for the helpful comments.

Response:

The title, introduction and abstract were modified during a first submission of the paper where they suggested changes to improve the manuscript.

The aim of the work is to use laboratory medicine as the new frontier for monitoring the health status of the competitive athlete.

The aims are also reiterated in the first part of the discussion to help the reader focus his attention on the purpose of the manuscript.

- Previous study showed an association between weather seasonality and blood parameters. As this study last for 3 months (possible weather seasonality change) the use of a control group is necessary to control this confounding variable.

We modified the abstract lines 35-40 to emphasize that through the use of laboratory biochemistry, we monitored several serological biomarkers to evaluate the health status of the 12 competitive athletes.

Introduction

- The rational of this study need to be more clarified. Indeed, this study focused on the chronic effect of training program on a number of biomarkers. However, the majority of cited studies in the introduction were dealing with the acute effect of physical exercises. It would be more appropriate to include previous studies dealing with physiological adaptation to different form of chronic exercises or training program or adaptation in different athletic populations (soccer, individual sports etc.) The rational of using basketball players should also be clear in the introduction.

- The authors have identified a number of biomarkers that can help monitoring health status in athletes as it was previously been showed to be altered during exercises. However, it should be noted that in the majority of these study the post exercise levels did not exceed the normal range in healthy individual and that levels alteration was explained by water loss especially during prolonged exercise.

- The authors indicate in the title and through the manuscript “Athletes’ passport” and they selected some biomarkers to create this passport. However, such biological passports cannot be limited to these biomarkers. A simple example is related to the inclusion of CRP (used by previous study as inflammation biomarker). Many studies showed that CRP level did not significantly change during exercise. It is also suggested to control Hs-CRP with at least one or two myokines (e.g., IL-6) to assess inflammatory responses. Additionally, it is well known that prolonged or high-intensity exercise results in oxidative damage to macromolecules in both blood and skeletal muscle; with overtraining induced high production of ROS may negatively affect skeletal muscle contractile function and muscle fiber adaptation. So, when speaking about an athlete’s biological passports it seems also important to include biomarkers of redox status. Therefore, I suggest avoiding using athletic passport and to make the used vocabulary as simple as possible, especially that the used design and tested parameters did not reflect such title and purpose (Athlete's passport: prevention of infections, 3 inflammations, injuries and cardiovascular diseases)

Response:

Thanks to the Reviewer 3 for the helpful comments.

The novelty of our study consists of monitoring a team of athletes every two months for two consecutive sports sessions. This continuous monitoring has enabled us to identify the most significant variations of specific parameters and, above all, to promptly intervene in the case of abnormalities to prevent accidents and/or drop of shape.

The basic idea of our study is to identify the changes of specific biomarkers. For this reason, we have chosen to monitor a small group of athletes and, on their blood samples, we have applied a large panel of biochemical and haematological parameters. By monitoring these athletes and always respecting the same methodologies and withdrawal times, any changes in the parameters were easily identified.

Although the values ​​are normal, they undergo significant variations which must be considered as alarm bells. in fact, the aim is to monitor and prevent injuries and / or infections and / or disorders

We did not perform the IL-6 dosage because it has been shown that intense exercise induces a progressive increase in the aforementioned interleukin (Érica Cerqueira et al; 2019 doi: 10.3389/fphys.2019.01550).

It is clear that the evaluation of a panel of cytokines for the evaluation of an inflammatory state is an excellent idea, we appreciate the advice of the Reviewer 3 and, for our future studies, we take into consideration the idea to monitor IL-6.

- Please provide the expansion at the first occurrence of all abbreviation (e.g., MPV, PCT)

Thanks to the Reviewer 3 for the helpful comments.

Response:

In order to satisfy your request, we have now modified section 1 lines 144,147.

Methods

- L197 “blood samples from athletes took place in three different sessions: at 0 months in the preseason phase; 1 month after the start of the championship; 3 months after the start of the championship” what you mean by “0 months in the preseason” is this baseline values just after the long between seasons break? 

Thanks to the Reviewer 3 for the helpful comments.

Response:

0 months indicated as pre season, represents the month in which the athletes begin the athletic preparation, before starting the athletic preparation, they undergo a medical examination at the gonista, at the same time the samples were taken.

Please give more information about the basketball season (length, how many matches, duration between matches etc.) Please argue the choose of these 3 time points. In which micro-cycle did you collect blood (e.g., opening, developing, stabilization, checking, tapering, racing, recovery) and what was the load (volume, intensity) during 2-3days just before collecting blood?

Thanks to the Reviewer 3 for the helpful comments.

Response:

The methods for collecting the samples are specified in paragraph 2.3; in addition, paragraph 2.2 explains both the duration of the training and the characteristics and the duration of the game.

Indeed, collecting blood just the day after a match and collecting blood at the end of a recovery micro-cycle will blind the effect of training period (e.g., 3 months of training). Did the authors ensured similar condition (e.g., amount of physical activities, diet behaviors sleep quantity) during at least 72h preceding the blood sample in each blood sampling time-points?

Thanks to the Reviewer 3 for the helpful comments.

Response:

The athletes practiced the same lifestyle and the same diet, the details are in paragraphs 2.2 and 2.3. Furthermore, the collection was carried out after 72 hours of rest, in the morning, on an empty stomach and before any form of training.

- Blood parameters showed also to follow a circadian variation, please specify the time of day for all blood samples.

Thanks to the Reviewer 3 for the helpful comments.

Response:

In paragraph 2.3 we highlight the pick-up time.

- L193: “The athletes included in the study had no previous muscle injuries, cardiovascular disorders, or 194 recurring infections” what about during the experimental period?

Thanks to the Reviewer 3 for the helpful comments.

Response:

Monitoring through laboratory doctors during these 3 months has allowed to prevent injuries and / or infections, in fact in subjects in which certain parameters were altered it was administered or drug therapy or an ad personam recovery program was carried out.

- About participants; please specify the gender, years of experiences, and the sample size calculation.

Thanks to the Reviewer 3 for the helpful comments.

Response:

We have modified paragraph 2.2 line 197.

- Is there a standardized meal the day of each blood sample?

Thanks to the Reviewer 3 for the helpful comments.

Response

We have modified paragraph 2.2 and added the nutrition performed by the 12 athletes

- Blood sample (quantity of blood, position of athlete during the blood sample (e.g., sitting how many minutes he was sitting before blood sample) duplicate analysis or not?). Methods for each biomarker should be provided.

Thanks to the Reviewer 3 for the helpful comments.

Response

Each athlete was subjected to a blood sample, the waiting time was about 10 minutes. The measurement was performed in duplicate (paragraph 2.3)

the methods used are specified in paragraph 2.4

- Statistical analysis, how normality of the data has been tested?

Thanks to the Reviewer 3 for the helpful comments.

Response

The statistics were carried out on raw data; in the present case data were expressed as the mean ± standard deviation. Student's t-test made comparisons among groups. Values ​​of p <0.05 were considered significant.

No normalization was performed, as can be seen from the graphs, since in a previous submission we were suggested to use raw data. we then performed the statistics of 0 months versus 1 and 3, and 1 month versus 3.

We have specified everything in paragraph 2.5 and in the legends

- Blood was taken in 3 time points, why the data was not analyzed using the one way Anova for repeated measures and post-hoc test (of course if it is normally distributed) ?

Thanks to the Reviewer 3 for the helpful comments.

Response

We preferred to use a non-parametric test such as t-Student, because there are three independent samples (0,1 and 3 months); this type of analysis seemed more appropriate. Indeed, the t-Student Test for

it is used for independent samples in order to compare the averages of two independent samples.

- Effect size (e.g., d cohen) should be provide with p values.

Thanks to the Reviewer 3 for the helpful comments.

Response

In each legend, the p values ​​attributed to the significance values ​​obtained are reported.

Results

- When stating “To determine how intense exercise” the reader will think that you are evaluation the acute response following an intense exercise. Therefore, it should be clear through the manuscript that this study evaluates the effect of chronic (or 3 months) basketball training program on some blood parameters.

Thanks to the Reviewer 3 for the helpful comments.

Response

With this expression we tend to emphasize that the athletes analyzed are professional athletes, and that the training, although performed under monitoring of the athletic trainers, inlunza the barameters considered in the study.

- Table 1. It is not clear if this table is dedicated to present the correlation between cortisol and fT3, Cortisol and fT4 and Cortisol and TSH at the different time point or this table showed the ration values (e.g., Cortisol/fT3 etc.)

Thanks to the Reviewer 3 for the helpful comments.

Response

The table shows the values ​​obtained at 0,1 and 3 months for each ratio. In order to evaluate this correlation over the 3 months.

- Exact p values with effect size (d cohen) should be presented in text when analyzing the differences in mean between time points.

Thanks to the Reviewer 3 for the helpful comments.

Response

In the legend we have reported the values ​​of each single p value

Discussion

- L320 “however, we have noticed a significant decrease in the average corpuscular volume (MCV). This data allows us to sustain that in our case an intense physical exercise can cause the appearance of anemia” before taking such conclusion, it should be clarified for all parameters values in all time point if these values are still in the normal (healthy) range or not.

Thanks to the Reviewer 3 for the helpful comments.

Response

The parameters obtained are normal, however a decrease over time can presume an probability of anemia appearing, in fact in the discussion from line 332 to line 339 we emphasize that monitoring is necessary to prevent the appearance of anemia.

Please avoid speaking about the effect of an intense physical exercise. As I mentioned above, we speak about the effect of intense exercise when comparing pre-exercise with post-exercise values (acute effect). However, in the present study you are collecting blood just before exercises in different time point of the seasons. So, you are looking for adaptation in different season period or effect of chronic or regular basketball training program (3months).

Thanks to the Reviewer 3 for the helpful comments.

Response

In paragraph 4 (discussion) from lines 321 to 327 we clarify that we are talking about adaptation.

- Please go deeper and try to explain to underlying mechanisms behind the present results and also try to link the results with the specificity of the training program in Basketball. Comparing the observed adaptation following basketball training program with other previously studied discipline will strengthen the discussion.

Thanks to the Reviewer 3 for the helpful comments.

Response

In paragraph 4 (discussion) from lines 382 to 396

- What about the results of the correlation, they are absent in the discussion.

Thanks to the Reviewer 3 for the helpful comments.

Response

In paragraph 4 (discussion) from lines 373 to 381 we discussed table 1 and then the correlations.

Reviewer 4 Report

This manuscript presents an interesting topic and is presented with rigor.

Introduction section is clear and easy to follow providing valuable information about the studied variables.

Methods.

In 2.1 paragraph there is no registration number, was this clinical trial registered in clinicaltrials.gov or similar?

I wonder if there were any inclusion criteria for the study? Maybe professional basketball players are a little bit ambiguous.

How was the sample selected? By convenience? Was any participant discharged before the beginning of the study?

Discussion.

This section is well structured and the obtained results are described and argued in detail. Nevertheless, there is some suggestion in this section.

The major concern with this section is that clinical application should be addressed with further detail. The correlation is well explained but how to determine the increase of risk of injury would enhance the usefulness of this interesting manuscript. Would it be possible to set a cut-off score regarding the studied parameters? This way the “alarm effect” of this test will help sports doctors to prevent injury during the competition as well as help sport coaches to manage training load during the competitive season.

Consider replacing the first paragraph of this section, which is quite similar to the introduction section statement.

“Following the observation of the results obtained, we observed” replace observation or observed in this phrase.

Author Response

Reviewer 4

This manuscript presents an interesting topic and is presented with rigor.

Introduction section is clear and easy to follow providing valuable information about the studied variables.

 Methods.

In 2.1 paragraph there is no registration number, was this clinical trial registered in clinicaltrials.gov or similar?

Thanks to the Reviewer 2 for the helpful comments.

Response:

Paragraph 2.1 shows the protocol number (protocol 200/17) approved by Ethics of the School of Medicine, University of Naples Federico II. The study is not a clinical trial, but rather represents a launching pad to assess the health of the athlete not only through the competitive medical examination.

I wonder if there were any inclusion criteria for the study? Maybe professional basketball players are a little bit ambiguous.

Thanks to the Reviewer 2 for the helpful comments.

Response:

The characteristics of the athletes examined are reported in paragraph 2.2. In addition, the 12 competitive athletes are part of the same team.

How was the sample selected? By convenience? Was any participant discharged before the beginning of the study?

Thanks to the Reviewer 2 for the helpful comments.

Response:

The selection criteria are shown in paragraph 2.2

No athlete was rejected, all 12 competitive athletes belonging to the same team after the competitive visit were selected for the experimental design, all 12 fell within the criteria reported in paragraph 2.2.

Discussion.

This section is well structured and the obtained results are described and argued in detail. Nevertheless, there is some suggestion in this section.

The major concern with this section is that clinical application should be addressed with further detail. The correlation is well explained but how to determine the increase of risk of injury would enhance the usefulness of this interesting manuscript. Would it be possible to set a cut-off score regarding the studied parameters? This way the “alarm effect” of this test will help sports doctors to prevent injury during the competition as well as help sport coaches to manage training load during the competitive season.

Consider replacing the first paragraph of this section, which is quite similar to the introduction section statement.

“Following the observation of the results obtained, we observed” replace observation or observed in this phrase.

Thanks to the Reviewer 2 for the helpful comments.

Response

The discussion aims to underline how the use of laboratory medicine can support the traditional athletic medical examination, in order to evaluate the athlete's health at 360 degrees.

In addition, monitoring every three months represents a new way to check the athlete's health without being aggravated where there is the presence of pathologies. This type of monitoring therefore helps both the athlete and sports doctors who can calibrate training and recovery from injuries ad personam.

The limits of the parameters considered are set for each parameter by the National guidelines, to which the Biochemistry Laboratory of Policlinico II of Naples, University of Naples Federico II refers.

Round 2

Reviewer 1 Report

The novelty of this study remains completely absent.
The monitoring of standard laboratory analysis as a general approach to correct deficiency conditions and prevent athletes injuries does not put in evidence any new clinical strategy or specific improvement.
The n of this study is too low.

Author Response

Reviewer 1

Comments and Suggestions for Authors

The novelty of this study remains completely absent.

The monitoring of standard laboratory analysis as a general approach to correct deficiency conditions and prevent athletes injuries does not put in evidence any new clinical strategy or specific improvement.

The n of this study is too low.

Response:

Thanks for the comments, in order to clarify our experimental approach and the novelty of our study, we would like to specify the following points:

1) All athletes engaged in competitive activities must undergo a medical examination after issuing a fitness certificate.

2) The protection of athletes in the medical field health is regulated by the Ministerial Decree of 18 February 1982. For recognition of fitness, athletes undergo the following tests:

  1. a) Medical examination (anamnesis of the athlete)
  2. b) urine test;
  3. c) electrocardiogram.

However, the clinical evaluation of the athlete, although well regulated, does not make use of tests

diagnostics aimed at monitoring biochemical parameters.

Therefore, the novelty lies in the monitoring of the athletes over time, as a support to the sports doctor to have a unitary picture of the health of the individual athlete.

Secondly, the number of athletes refers to the monitored basketball team, which is why no other athletes have been added to have a homogeneous sampling of athletes who are placed in the same conditions.

Reviewer 2 Report

Unfortunately, the authors did not respond to some of the comments and did only minor adjustments. Major problems, such as the inappropriate and rather superficial discussion of the study results, have still not been addressed. This is rather disappointing, as it suggests lack of seriousness by the authors. Studies should meet a certain minimum standard of good scientific practice, including critical analysis of the results. This clearly did not happen in this case, the authors even draw false conclusions.

Author Response

Reviewer 2

Comments and Suggestions for Authors

Unfortunately, the authors did not respond to some of the comments and did only minor adjustments. Major problems, such as the inappropriate and rather superficial discussion of the study results, have still not been addressed. This is rather disappointing, as it suggests lack of seriousness by the authors. Studies should meet a certain minimum standard of good scientific practice, including critical analysis of the results. This clearly did not happen in this case, the authors even draw false conclusions.

Response

Thanks for the comments and suggestions.

First of all, we want to point out firmly that our seriousness should not be discussed at all, nor should allegations regarding false conclusions be affirmed.

Regarding the results, the graphs have been modified as suggested by the Editor; also, the results obtained were critically analyzed in the discussion, without reporting any false news, instead of trying to evaluate the data collected at 360 degrees, underlining that some of the parameters assessed could be the alarm bell for any diseases to investigate.

Reviewer 3 Report

Unfortunately, it seems that the authors was not serious in their revisions. Although, I tried to to give many suggestions and possible corrections to improve the quality of the papers and show the rational for this study. The authors have only made small modifications in some parts of the manuscript with no change in the introductions.

Author Response

Reviewer 3

Comments and Suggestions for Authors

Unfortunately, it seems that the authors was not serious in their revisions. Although, I tried to to give many suggestions and possible corrections to improve the quality of the papers and show the rational for this study. The authors have only made small modifications in some parts of the manuscript with no change in the introductions.

Response

Thanks for the comments and suggestions.

First of all, we want to point out that our seriousness is an indisputable principle. We improved the quality of the paper according to your suggestions: to satisfy your request, we have changed part of the abstract and also the introduction has been modified and reduced. The changes made are highlighted in yellow.